

# Validation of three functional tests against the 6-minute walk test in individuals with obesity

Thapanun Mahisanun[1] and Jittima Saengsuwan[2]

[1] Department of Rehabilitation Medicine, Thabo Crown Prince Hospital, Nong Khai, Thailand
[2] Department of Rehabilitation Medicine, Faculty of Medicine, Khon Kaen University, Khon Kaen, Thailand

## ABSTRACT

**Background**. The six-minute walk test (6MWT) is one of the most commonly-used assessments of functional capacity. However, its length and space requirements can limit feasibility in busy clinical settings. There is limited evidence on the validity of shorter or alternative functional tests compared to the 6MWT in individuals with obesity. Therefore, this study aimed to evaluate the concurrent validity of the distance covered during the first two minutes of the 6-minute walk test (2'6MWT), the 2-minute step test (2MST) and the 1-minute sit-to-stand test (1STST) against the full 6MWT in individuals with obesity.

**Methods**. This was a cross-sectional study. Participants with obesity completed three functional tests—6MWT, 2MST and 1STST—on the same day. Spearman's rank correlation was used to assess the concurrent validity of the 2'6MWT, 2MST and 1STST in relation to the 6MWT.

**Results**. A total of 59 individuals with obesity (16 men) participated in the study. Age was 34 (8.5) years (mean (standard deviation)), and body mass index was 40.9 (6.5) $kg/m^2$. The 2'6MWT demonstrated a strong correlation with the 6MWT (Spearman's rank correlation, $r_s = 0.87$, $p < 0.001$), while the 2MST and the 1STST showed moderate and weak correlations, respectively ($r_s = 0.45$, $p < 0.001$; $r_s = 0.31$, $p = 0.02$).

**Conclusion**. The 2'6MWT may serve as a valid alternative to the 6MWT for assessing functional capacity in individuals with obesity, offering a shorter duration and potentially reducing fatigue. The 2MST may be a more suitable alternative to the 1STST in settings with limited space for walking tests.

Corresponding author
Jittima Saengsuwan,
sjittima@kku.ac.th

## INTRODUCTION

The escalating obesity epidemic is one of the most serious global health challenges (*Koliaki, Dalamaga & Liatis, 2023*). Projections indicate that by 2050, nearly 60% of adults will be classified as overweight or obese (*Ng et al., 2025*). Obesity is strongly associated with an increased risk of cardiovascular diseases, including atherosclerotic dyslipidemia, type 2 diabetes mellitus, and hypertension. Additionally, it promotes inflammation and oxidative stress, further increasing the risk of coronary artery disease, heart failure or stroke (*Lopez-Jimenez et al., 2022*). Obesity is often linked to poor diet quality, reduced physical activity

and sedentary behaviour (*Alosaimi et al., 2023*; *Lopez-Jimenez et al., 2022*; *Neta et al., 2024*). Consequently, both diet and physical activity play a crucial role in interdisciplinary obesity management. The benefits of physical activity for weight management, body composition, physical fitness and cardiometabolic health in individuals with obesity are well-documented (*Oppert, Bellicha & Ciangura, 2021*; *Raiman et al., 2023*).

Evaluating functional capacity is crucial for individuals with obesity. Various tests have been used, but the 6-minute walk test (6MWT) is the most commonly utilized for assessing fitness (*ATS Committee on Proficiency Standards for Clinical Pulmonary Function Laboratories, 2002*). The 6MWT is a sub-maximal exercise test in which individuals are asked to walk for 6 min, and the distance covered is recorded. This test closely resembles daily activities, is simple to administer and is inexpensive (*ATS Committee on Proficiency Standards for Clinical Pulmonary Function Laboratories, 2002*). However, the 6MWT has certain limitations. The six-minute duration may not be practical in busy clinical settings, it can be exhausting for individuals with severe sedentary lifestyles, and it may cause discomfort for those experiencing joint pain. Additionally, the test requires a long corridor, which may not be available in all facilities.

To address these limitations, alternative functional tests, such as the 2-minute step test (2MST) or 1-minute sit to stand test (1STST), have been proposed. The 2MST requires individuals to lift each knee to the point midway between the patella and the iliac crest. Physical support is provided if balance issues arise. The test is scored by counting the number of times the right knee reaches the required height within two minutes (*Jones & Rikli, 2002*). The 1STST is more convenient and requires less time compared to other tests. In this test, the participant sits with arms crossed over the chest and repeatedly stands up fully and sits down as quickly as possible. The total number of complete sit-to-stand cycles performed within one minute is recorded (*Bohannon & Crouch, 2019*). Several studies have investigated the validity of these tests. *Fernandes et al. (2021)* and *Meriem et al. (2015)* found that 1STST was moderately correlated with the 6-minute walking distance (6MWD) in patients with chronic obstructive pulmonary disease ($r = 0.47$ to $0.54$). Additionally, *Bohannon et al. (2014)* analysed the distance covered during the first two minutes of the 6-minute walk test (2'6MWT) and found a strong correlation with the 6MWD in community-dwelling children and adults ($r = 0.97$). Since the distance covered in the 2'6MWT is comparable to that of the standalone 2MWT in clinical populations (*Valet, Pierchon & Lejeune, 2023*), we employed 2'6MWT as a plausible and valid proxy.

In individuals with obesity, limited studies have examined the validity of alternative functional tests in comparison to the 6MWT. In our clinical settings, the use of the 6MWT is restricted due to space and time constraints. Therefore, we aimed to identify a valid alternative. Since validity is population-specific, our primary objective was to assess the concurrent validity of the distance covered during the first two minutes of the 6MWT (*i.e.,* 2'6MWT), 2-minute step test (2MST), and the 1-minute sit-to-stand test (1STST) in relation to the full 6-minute walk test (6MWT) in individuals with obesity.

## MATERIAL & METHODS

### Participants

A cross-sectional descriptive study was conducted on patients with obesity attending the obesity clinic at Thabo Crown Prince Hospital, Nong Khai, Thailand, between November 2024 and January 2025. Participants were recruited according to the following inclusion criteria: (1) age between 18 and 60 years, (2) BMI $\geq$ 30 kg/m$^2$, and (3) ability to communicate. Exclusion criteria included patients who had undergone surgery or experienced severe joint pain that impaired lower extremity movement, those with severe cardiovascular or pulmonary diseases, and individuals with severe hearing, visual or sensory impairments. All participants provided written informed consent prior to their inclusion in the study. The study was conducted in accordance with the principles of the Declaration of Helsinki and was approved by the Institutional Review Board of the Nong Khai Provincial Public Health Office (IRB number: 88/2567).

### The distance covered during the first two minutes of the 6-minute walk test and the 6-minute walk test

The 6MWT was conduct in accordance with the American Thoracic Society guidelines (*ATS Committee on Proficiency Standards for Clinical Pulmonary Function Laboratories, 2002*) on a 15 m corridor. During the test, participants walked around two cones for six minutes. If they felt breathless or exhausted, they were allowed to slow down, stop, or sit and rest. The distance covered during the first two minutes (2'6MWT) and after the full six minutes (6MWT) was recorded. The two-minute distance was derived from continuous walking during the 6MWT without pausing the test.

### 2-minute step test

The researcher first determined the required knee-raising height by measuring the midpoint between the top of the patella and the iliac crest. The tape marker was then placed on the wall. The participant stood with their feet shoulder-width apart. Upon the researcher's signal, the participant alternately raised their knees as quickly as possible and continued for two minutes (*Poncumhak et al., 2023*). One full cycle consisted of raising both the left and right knees. If the participant was unable to reach the target height, they were allowed to slow down or stop until they could achieve the required height.

### 1-minute sit-to-stand test

A standard chair (46 cm in height) without armrests was placed against a wall. The participant sat upright with their knees and hips at a 90-degree angle, feet flat on the ground and hip-width apart, with arms crossed over their chest. They were instructed to stand up fully and sit back down in the same position, repeating as many times as possible within one minute (*Bohannon & Crouch, 2019*). If the participant felt tired, they were allowed to rest during the one-minute period.

### Study protocol

After obtaining informed consent, baseline demographic and anthropometric data (height, weight, and waist circumference), as well as baseline heart rate, blood pressure, and

oxygen saturation were collected. The level of physical activity was assessed using the questionnaire from the Aerobics Center Longitudinal Study (*Stofan et al., 1998*). Each participant sequentially performed the 6MWT, 2MST and 1STST, with a 15-minute interval between each test. Heart rate, blood pressure and blood oxygen saturation were measured before and after each test. Additionally, breathlessness and leg fatigue were assessed using the Borg CR scale (R) (CR10) (*Borg, 1998*; *Borg, 2004*; *Borg, 1982*). The authors have permission to use this instrument from the copyright holders.

## Sample size calculation

The sample size was calculated using the formula provided by *Hulley et al. (2013)*. Based on an estimated correlation coefficient of 0.40, with a significant level of alpha = 0.05 and 80% power, a total of 47 participants was required. After adjusting to a 20% dropout rate, the final required sample size was 59 participants.

## Statistical analysis

Quantitative data are presented as mean and standard deviation for normally distributed data, while median and interquartile range are used for non-normally distributed data. Categorical data are described using frequency and percentage.

Since the data were not normally distributed, Spearman's rank test was used to assess concurrent validity. The correlation coefficient ($r_s$) was interpreted as follows: 0.0–0.10, negligible correlation; 0.10–0.39, weak correlation; 0.40–0.69, moderate correlation; 0.70–0.89, strong correlation; 0.90–1.00, very strong correlation (*Schober, Boer & Schwarte, 2018*). A *p*-value of < 0.05 was considered statistically significant. All statistical analyses were performed using Stata (Stata Statistical Software: Release 18. College Station, TX, USA: Stata Corp LLC).

## RESULTS

Fifty-nine participants (43 women, 16 men) were enrolled and completed the study. Their average age was 34 years, and their average BMI was 40.9 kg/m$^2$. The majority (62.3%) had a low level of physical activity (Table 1).

All patients successfully completed the tests. The median 6-minute walk distance (6MWD) was 452.0 m. The median distance for the distance covered during the first 2 min of the 6MWT (2'6MWT) was 153.5 m while the mean number of steps in the 2-minute step test (2MST) was 74.3. The median number of repetitions in the 1-minute sit-to-stand test (1STST) was 31 (Table 2).

The physiological responses to functional tests (heart rate, blood oxygen saturation (SpO$_2$) and the Borg CR10 scale) are shown in Table 2. Significant differences were observed in heart rate and Borg CR10 scale ratings for breathlessness and leg fatigue across all tests. The highest median heart rate and Borg CR10 scale for dyspnea and leg fatigue were observed in the 2MST (Table 2). Walking speed during the 2'6MWT and the last four minutes of the 6MWT were comparable, with a mean gait speed of 1.28 m/s in both tests ($p = 0.71$).

Scatter plots of 2'6MWT, 2MST, 1STST and the 6MWT are shown in Fig. 1. The 2'6MWT demonstrated a strong correlation with the 6MWT ($r_s = 0.87$) while the 2MST

**Table 1 Demographic data and clinical characteristics of the participants (N = 59).**

| Variables | N (%) |
|---|---|
| Male | 16 (27.1) |
| Age (year), mean (SD) | 34 (8.5) |
| Body mass index (kg/m²), mean (SD) | 40.9 (6.5) |
| Waist circumference (cm), mean (SD) | 112.6 (15.9) |
| Smoking | 2 (3.4) |
| Alcoholic drinking | 20 (33.9) |
| Underlying disease | 22 (37.3) |
|     Hypertension | 11 (18.6) |
|     Diabetes Mellitus type 2 | 11 (18.6) |
|     Dyslipidemia | 4 (6.9) |
| Physical activity[*] | |
|     No activity | 6 (10.2) |
|     Participated in sporting or leisure-time physical activity other than walking, jogging, or running | 37 (62.7) |
|     Walk, jog, or run up to 10 miles per week | 5 (8.5) |
|     Walk, jog, or run from 10 to 20 miles per week | 11 (18.4) |
| Laboratory results, mean (SD) (n = 45) | |
|     Fasting blood sugar (mg/dl) | 116.5 (59.0) |
|     Cholesterol (mg/dl) | 194.2 (30.1) |
|     Low density lipoprotein (mg/dl) | 119.4 (28.4) |
|     High density lipoprotein (mg/dl) | 44.9 (8.6) |
|     Triglyceride (mg/dl) | 160.7 (100.1) |

**Notes.**

Data are presented in n (%) unless otherwise specified.

*Category based on Aerobics Center Longitudinal Study (*Stofan et al., 1998*).

and the 1STST showed moderate and weak correlation with the 6MWT ($r_s$ = 0.45 and 0.31, respectively) (Fig. 1).

# DISCUSSION

The aim of this study was to determine the concurrent validity of the distance covered during the first two minutes of the 2'6MWT, 2MST and the 1STST in comparison with the full 6MWT in individuals with obesity. The 1STST showed weak correlation with the 6MWT ($r_s$ = 0.31), while the 2MST demonstrated moderate correlation ($r_s$ = 0.45). In contrast, the 2'6MWT exhibited strong correlation with the full 6MWT ($r_s$ = 0.87).

Compared to findings in other populations, the correlations for the 2MST and 1STST were generally lower in our cohort. For 1STST, the concurrent validity of the 1STST with the 6MWT was 0.54 in healthy participants (*Ozalevli et al., 2007*), 0.57 to 0.75 in patients with COPD (*Ozalevli et al., 2007*; *Vaidya et al., 2016*), 0.54 in post-coronary artery bypass graft patients (*Anukul et al., 2024*), and 0.75 in patients with COVID-19 (*Honpode et al., 2021*). Similarly, the 2MST-6MWT correlation ($r_s$ = 0.45) was lower than that seen in elderly patients with hypertension (0.75) (*Poncumhak et al., 2023*) and in individuals with lumbar spinal stenosis (r = 0.82) (*Paker et al., 2025*).

**Table 2  Results and changes in cardiorespiratory parameters during the 2'6 MWT, 6 MWT, 2 MST, and 1STST.**

| Variables | Baseline Mean (SD) | End Mean (SD) | Changes from baseline MD (95% CI) | *P* value |
|---|---|---|---|---|
| The distance covered during the first 2 min of the 6MWT (2'6MWT) | | | | |
|   2-minute walk distance, median (IQR) | – | 153.5 (28) | – | – |
|   6-minute walk test (6MWT) | | | | |
| 6-minute walk distance, median (IQR) | – | 452.0 (72.3) | – | – |
|   Heart rate (bpm) | 87.6 (11.0) | 116.0 (12.9) | 28.5 (25.6 to 31.4) | <0.001 |
|   Borg scale CR10 dyspnea | 1.7 (1.6) | 5.0 (1.8) | 3.3 (2.8 to 3.7) | <0.001 |
|   Borg scale CR10 leg fatigue | 1.2 (1.4) | 4.8 (2.1) | 3.6 (3.0 to 4.2) | <0.001 |
|   Oxygen saturation (%) | 97.9 (1.0) | 97.7 (0.9) | −0.0 (−0.4 to 0.0) | 0.096 |
| 2-minute step test (2MST) | | | | |
|   Number of successive steps | – | 74.3 (13.1) | – | – |
|   Heart rate (bpm) | 89.0 (10.7) | 118.2 (13.4) | 29.2 (25.8 to 32.6) | <0.001 |
|   Borg scale CR10 dyspnea | 1.7 (1.6) | 5.4 (1.90) | 3.7 (3.2 to 4.3) | <0.001 |
|   Borg scale CR10 leg fatigue | 1.2 (1.4) | 5.1 (2.0) | 3.9 (3.3 to 4.4) | <0.001 |
|   Oxygen saturation (%) | 98.2 (1.0) | 98.0 (1.0) | −0.2 (−0.6 to 0.1) | 0.13 |
| 1-minute sit to stand test (1STST) | | | | |
|   Number of successive stands, median (p25, p75) | – | 31 (25, 37) | – | – |
|   Heart rate (bpm) | 88.3 (10.7) | 114.7 (13.3) | 26.4 (23.2 to 29.6) | <0.001 |
|   Borg scale CR10 dyspnea | 1.7 (1.6) | 5.1 (2.1) | 3.4 (2.9 to 3.8) | <0.001 |
|   Borg scale CR10 leg fatigue | 1.2 (1.4) | 4.4 (2.4) | 3.2 (2.6 to 3.8) | <0.001 |
|   Oxygen saturation (%) | 98.1 (0.9) | 97.7 (1.3) | −0.4 (−0.7 to 0.2) | 0.002 |

**Notes.**

Abbreviations: CI, confidence interval; IQR, Interquartile range; MD, mean difference; SD, standard deviation.

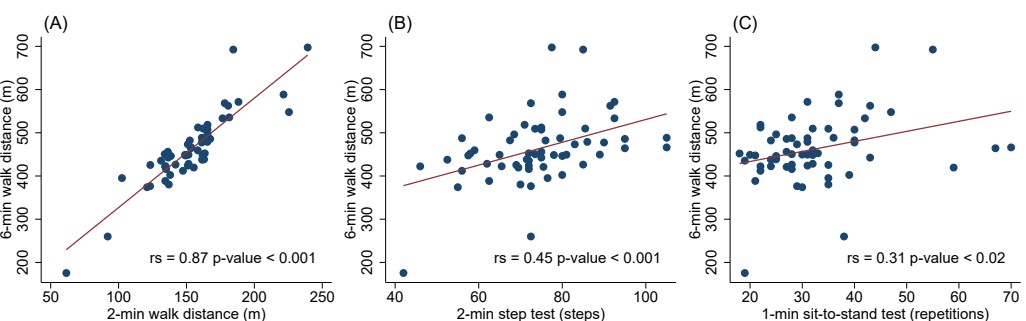

**Figure 1  Scatter plots showing the correlations between functional tests and the 6-minute walk test (6MWT).** (A) 2'6MWT *vs.* 6MWT, (B) 2MST *vs.* 6MWT, and (C) 1STST *vs.* 6MWT.

The 2MST and 1STST results were lower in individuals with obesity compared to other populations. In healthy Indian women aged 30 to 39 years, the average number of steps in the 2MST was 86.8 (*Lohakare & Jiandani, 2023*), whereas our study found a median of 74.3 steps. Similarly, the reference value for the 1STST for the Portuguese women of the same age group was 41 repetitions (*Vilarinho et al., 2024*). Although comparisons across

studies should be interpreted with caution—given potential differences in functional ability among various ethnic groups and countries (*Nguyen et al., 2024*)—the lower performance observed in our study may be partly attributed to the challenges in performing 2MST and 1STST in individuals with obesity. During the 1STST, over 30% of participants experienced difficulty completing the movement. Most were able to maintain a consistent pace during the first 30 s; however, in the second half of the test, they slowed down and required breaks. Participants reported knee pain and discomfort due to friction between their thighs. In the 2MST, more than 10% of participants struggled to raise their legs to the required height. This difficulty was particularly common in those with substantial subcutaneous fat accumulation in the thigh area. To compensate, the participants leaned backwards to lift their legs higher, leading to an unstable posture during the test. In addition, the low correlation may reflect participants' unfamiliarity with performing these otherwise routine movements repeatedly and at a high intensity.

The distance covered in the 2'6MWT was strongly correlated with the 6MWT ($r_s = 0.87$). This finding is comparable to previous studies which reported a correlation of 0.97 in community dwelling adults and children (*Bohannon et al., 2014*) and 0.99 in individuals with subacute or chronic stroke (*Valet, Pierchon & Lejeune, 2023*). Compared to individuals with normal weight, we found comparable distances in both the 2'6MWT and 6MWT among individuals with obesity (153.5 and 452.0 m, respectively) and in Vietnamese women aged 30 to 39 years, the mean distances for the two-minute walk test (2MWT) and 6MWT were 153 m and 444 m, respectively (*Nguyen et al., 2024*). This suggests that walking-based tests may be more suitable for individuals with obesity, as walking exerts less force on the joints and requires less balance and coordination than step-based or sit-to-stand activity. Additionally, the walking speed of individuals with obesity was consistent between the first 2'6MWT of the 6MWT and the last four minutes. *Valet, Pierchon & Lejeune (2023)* Also found that the difference in distance covered between the 2MWT and 2'6MWT was approximately 5 m, which appears negligible, suggesting that both tests demonstrated similar predictive performance for the 6MWT in individuals with subacute to chronic stroke. These findings suggest that both the 2'6MWT and 2MWT may serve as practical alternatives to the 6MWT in time-constrained situations. It should also be noted that the 6MWT may be more suitable for capturing cardiovascular and muscular endurance, as well as fatigue or decline in pace over time. In contrast, the 2'6MWT and other functional tests used in our study primarily reflect short-term functional capacity.

Our study has some limitations. The length of our walkway for the 6MWT was restricted to 15 m due to corridor space constraints. Although 15 m walkways are commonly used in hospital settings (*Cheng et al., 2020*; *Krasny, Jozwiak & Rodby-Bousquet, 2023*), the total distance covered in the 6MWT may be reduced compared to the recommended 20 m, 30 m or 50 m corridors, as more time is required for participants to turn (*ATS Committee on Proficiency Standards for Clinical Pulmonary Function Laboratories, 2002*). For example, studies have shown that individuals with chronic lung diseases and children with cerebral palsy tend to walk longer distances on a 30 m course compared to a 15 m course, likely due to fewer interruptions for turning (*Gochicoa-Rangel et al., 2020*; *Krasny, Jozwiak & Rodby-Bousquet, 2023*). While the minimal difference in distance

between the 2MWT and 2'6MWT supports the potential of the 2'6MWT as a proxy measure, this evidence is currently limited to individuals with chronic stroke. These findings should be interpreted with caution when generalizing to individuals with obesity or other populations. The functional tests were administered in a fixed sequence rather than being randomized or counterbalanced. This could have introduced order effects, including learning or fatigue bias, which may have influenced performance on subsequent tests. Future studies should consider randomizing or counterbalancing the order of tests as well as testing on separate days to minimize such effects. Additionally, as this was a single-centre study, the generalizability of our findings may be limited. While we focused on assessing the validity of the functional tests through correlation analysis, we did not include measures of reliability as each test was performed only once. Incorporating reliability assessments would have offered a more comprehensive psychometric evaluation.

In summary, the 2'6MWT demonstrated a strong correlation with the 6MWT, while the 2MST showed a moderate correlation and the 1STST showed a weak correlation. This may be attributed to the similarity between the 2'6MWT and 6MWT, as well as the physical challenges individuals with obesity face when performing the 2MST and 1STST. Therefore, the 2'6MWT or 2MWT appear to be a more suitable alternative to the 6MWT. In cases where space is limited, the 2MST may be a preferable alternative to the 1STST.

## CONCLUSIONS

The 2'6MWT may serve as an alternative to the 6MWT for measuring functional capacity in individuals with obesity, requiring less time and potentially causing less fatigue. In situations where space is limited for a walking test, the 2MST may be a more suitable alternative to the 1STST.

## ACKNOWLEDGEMENTS

During the preparation of this work, AI tools were used to improve the readability and language of the manuscript. The authors subsequently revised and edited the content as necessary, taking full responsibility for the final version of the manuscript.

### Funding
The authors received no funding for this work.

### Competing Interests
The authors declare there are no competing interests.

### Author Contributions
- Thapanun Mahisanun conceived and designed the experiments, performed the experiments, analyzed the data, prepared figures and/or tables, authored or reviewed drafts of the article, and approved the final draft.

- Jittima Saengsuwan conceived and designed the experiments, analyzed the data, prepared figures and/or tables, authored or reviewed drafts of the article, and approved the final draft.

## Ethics

The following information was supplied relating to ethical approvals (i.e., approving body and any reference numbers):

Ethical approval for this study was granted by the Institutional Review Board of the Nong Khai Provincial Public Health Office (IRB number: 88/2567). All participants provided written informed consent before being included in the study.

## Data Availability

The raw data are available in the Supplementary Files.

## Supplemental Information

Supplemental information for this article can be found online at http://dx.doi.org/10.7717/peerj.19755#supplemental-information.

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
