# Peer review of "Validation of three functional tests against the 6-minute walk test in individuals with obesity"

_PeerJ, doi:10.7717/peerj.19755_

## Round 0.1 · original submission · Major Revisions

Reviewer 1 ·

Basic reporting

1.Please use references within the last 10 years

2.Structure and Writing (Grammar and academic style)
A sentence such as “Participants walked continuously for the entire six minutes, without stopping to measure the two-minute distance separately.” should be refined to sound more scientific. For example: “The two-minute distance was derived from continuous walking during the 6MWT without pausing the test.”
There is a repetition in the statistics section: the sentence “Since the data were not normally distributed...” is written twice.

Experimental design

1. Methodology
The single-center design and use of a 15-meter corridor may introduce systematic bias, as it can underestimate the total distance compared to standard corridors of 30–50 meters. Although the authors acknowledge this as a limitation, it would be beneficial to include references that highlight its impact on the 6MWD (e.g., "participants typically lose 15–30 meters in total distance due to frequent turns").
The order of functional tests was not randomized. This could introduce an order effect or fatigue bias. A randomized or counterbalanced test sequence across participants would have been preferable.

2. Data Interpretation
The low correlations of the 2MST and 1STST were attributed to the unsuitability of these tests for individuals with obesity. However, the possibility that participants’ lack of experience or familiarity with these tests may have influenced the results was not discussed. Consider adding a sentence such as: “The low correlation may also reflect unfamiliarity with these test procedures among participants.”
The Borg scale and heart rate (HR) data were not explicitly discussed in the discussion section. These findings could be used to support the claim that the 2’6MWT induces less fatigue compared to the full 6MWT.

Validity of the findings

no comment

Additional comments

Figure 1 is not available in the provided manuscript. Please attach the figure or include a visual placeholder. Tables 1 and 2 are important and are referenced in the results section. Kindly include the supporting files and present the corresponding data.

Annotated reviews are not available for download in order to protect the identity of reviewers who chose to remain anonymous.

Reviewer 2 ·

Basic reporting

Thank you for the opportunity to review this manuscript entitled ‘Validation of three functional tests against the 6-minute walk test in individuals with obesity’ by Mahisanun and Saengsuwan. In the present study, authors tried to examine the validity of three functional tests in 59 adults with obesity. I think that the aim of this study is not so relevant since the 6MWT is already a valid and well-tolerated instrument to assess walking capacity in adults with type I and II obesity. I think that this article needs to be rejected due to the theme and also because of some methodological doubts. Below you will find my comments.

Abstract: In my opinion the abstract misses a background on why choosing to use the first two minutes of the 6MWT. I suggest rewriting the background, adding what is already known and why your article is needed. Moreover, I suggest summarizing the results section, maintaining only the most relevant results.

Introduction: I think that the introduction has some flaws. For example, I think it is not explained why you have chosen the first 2 minutes of the 6MWT and not performing a 2MWT. Moreover, I think that the references need to be updated, for example in the prevalence you should refer to the new prevalence published in 2025 in The Lancet (https://www.thelancet.com/journals/lancet/article/PIIS0140-6736(25)00397-6/fulltext).

Experimental design

Materials and methods: I think that even if the study is simple the methods are well-explained. However, I suggest a few corrections. Firstly, have you followed the declaration of Helsinki when performing your study? If yes, please state this. Then, regarding the sample size estimation, why have you chosen 0.40 as r of correlation? Why did you not consider including a simple blood sample analysis to understand what test was better related with the health of adults with obesity?

Validity of the findings

Results: I think that this section is well written, however I think that to be complete you need to present bland-altman plot (for agreement) and test-retest reliability (intraclass correlation coefficients).

Discussion: The discussion is adequately organized but lacks depth. While the authors correctly interpret their results, they overstate the importance of finding a strong correlation between a shorter and longer version of an already validated test. They mostly confirm existing knowledge without offering substantial new insights. Major limitations, such as the incomplete nature of the validation (relying only on correlation without assessing agreement or test-retest reliability), are briefly mentioned but not critically discussed. Overall, the discussion reads as an exercise in justifying a study that adds limited value to the existing literature.

Reviewer 3 ·

Basic reporting

Basic reporting was quite clear. Please note lines where content was repeated and requires editing noted below in Additional Comments.

Experimental design

Experimental design had some important limitations, such as
- an order for the three tests (even though 15min rest in between), and
- reporting on the initial two min of the six min test as the two minute test, rather than a separate test which might influence self-paced performance.

Validity of the findings

There are no concerns about validity.

Additional comments

Thank you for completing this study. I hope that the following recommendations can improve expression of the current study and considerations for future studies.


Tests performed in a specific order with 6min first presents a substantial risk of order effects such as incremental fatigue affecting the comparison. The weakest correlation was between the first and last tests. It is also noted that 30% had difficulty completing the final 1STST, where none had difficulty with the other tests. The potential for some findings to represent an order effect is a real limitation. Testing on separate days with a random order of tests could be of benefit if there is a risk of fatigue variably affecting performance.

Please explain why ¾ female? This is also a limitation as 16 is only a modest number of males.

Lines 70-71 and 77 are similar. Recommend removing the explanation at 70-71.

If the 2min recording was the initial two minutes of the 6MWT, this presents a methodological limitation. High correlation shows that people held their pace consistent. There is a possibility that if people are only attempting a 2min walk test then their pace may be greater than the initial 2 min of a 6min.

Results: where medians are reported, please also report interquartile ranges.

Discussion opening paragraph should answer the primary question, about correlations between the tests.

Line 223 appears to repeat line 218. Please remove one of these. It could be helpful to have separate paragraphs for the comparisons with 1STST and a paragraph for 2MST.

Considering strengths and weaknesses of the 2 and 6min tests, there is also the need to consider that they might reflect different ranges of capacity for walking in cohorts such as this with low cardiovascular fitness.

Please note the important limitation that if a 2min walk test were performed separately to the 6min walk test, then results may differ from those in the current data where the 2min was measured from the initial minutes of the 6min test.

Please include lines of best fit for the panels in Figure 1.

---

## Round 0.2 · accepted · Accept

The original Academic Editor is unavailable so I have reviewed the responses to the 3 Reviewers and am pleased to recommend the amended manuscript for publication. Thank you for selecting PeerJ, we look forward to future submissions. Thanks, A/Prof Mike Climstein

Reviewer 1 ·

Basic reporting

1.1. Language and Writing Style
The manuscript is written in clear, professional, and comprehensible English.
1.2. Background and References
The introduction provides a solid background, clearly contextualizing the importance of functional capacity testing in individuals with obesity. The references are current and relevant, covering both epidemiological data and prior validation studies.
1.3. Structure and Visual Data Presentation
The manuscript follows the standard scientific structure. Figures and tables are generally well presented
1.4. Raw Data
Raw data has been provided and appears complete in accordance with PeerJ’s policies.

Experimental design

2.1. Originality and Relevance
This is a relevant validation study with important clinical implications, especially for healthcare professionals working in primary care settings with limited space and time.
2.2. Clarity of Research Question
The research question is clearly defined: to evaluate the concurrent validity of the 2-minute distance from the 6MWT (2′6MWT), the 2-Minute Step Test (2MST), and the 1-Minute Sit-to-Stand Test (1STST) against the full 6MWT in individuals with obesity.
2.3. Methods and Ethics
The methodology is well described and follows ATS guidelines for the 6MWT. Use of standard tools (e.g., Borg scale) is appropriate, and ethical approval is clearly stated, including IRB number.
2.4. Limitations
The 6MWT was performed on a 15-meter course, which may affect walking distance results.
Suggestion: Consider further discussing how these limitations could affect generalizability.

Validity of the findings

3.1. Robustness and Consistency of Findings
The statistical analysis (Spearman correlation) is appropriate. Correlation values are interpreted using accepted classification standards. Findings are consistent with prior literature.
3.2. Conclusions
The conclusions are appropriately drawn from the results and are not overstated.
3.3. Additional Recommendations
The authors may consider noting that although a strong correlation was found between the 2′6MWT and the 6MWT, these tests measure slightly different durations and components of functional capacity (submaximal vs short-duration effort), and their use may depend on specific clinical goals.

Additional comments

This is a well-conducted validation study with high practical relevance for clinical use in obese populations.
Discuss more critically the limitation of using a 15-meter walkway for the 6MWT.
The physiological response assessment is comprehensive and the test procedures are appropriate.
Including reliability measures in future studies is recommended to strengthen the psychometric profile of the tests.

Annotated reviews are not available for download in order to protect the identity of reviewers who chose to remain anonymous.

Reviewer 3 ·

Basic reporting

no comment

Experimental design

no comment

Validity of the findings

no comment

Additional comments

All concerns were raised in the intial round of review. The author responses to reviewer comments were satisfactory in acknowledging limitations and directions for future research to sharpen the methodology. Hence I would support acceptance of the revised manuscript.